# Effect of Precipitates on the Mechanical Performance of 7005 Aluminum Alloy Plates

**DOI:** 10.3390/ma15175951

**Published:** 2022-08-28

**Authors:** Ni Tian, Xu Jiang, Yaozhong Zhang, Zijie Zeng, Tianshi Wang, Gang Zhao, Gaowu Qin

**Affiliations:** 1School of Materials Science & Engineering, Northeastern University, No. 3-11, Wenhua Road, Heping District, Shenyang 110819, China; 2Key Laboratory for Anisotropy and Texture of Materials (Ministry of Education), Northeastern University, No. 3-11, Wenhua Road, Heping District, Shenyang 110819, China

**Keywords:** 7005 aluminum alloy plate, precipitate, elongation, fatigue

## Abstract

In this study, the strength, elongation, and fatigue properties of 7005 aluminum alloy plates with different configurations of precipitates were investigated by means of tensile tests, fatigue tests, and microstructural observation. We found that the number and size of GP zones in an alloy plate matrix increased and the distribution was more uniform after the aging time was extended from 1 h to 4 h at 120 °C, which led to a rise in both strength and elongation of alloy plates with the extending aging time. The fatigue life of the alloy plates shortened slightly at first, then significantly prolonged, and then shortened again with the aging time extending from 1 h to 192 h and a fatigue stress level of 185 MPa and stress ratio (R) = 0. After aging at 120 °C for 96 h, the precipitates in the alloy plate matrix were almost all metastable η′-phase particles, which had the optimal aging strengthening effect on the alloy matrix, and the degree of mismatch between the α-Al matrix and second-phase particles was the smallest; the fatigue crack initiation and propagation resistances were the largest, leading to the best fatigue performance of alloy plates, and the fatigue life of the aluminum plate was the longest, up to 1.272 × 10^6^ cycles. When the aging time at 120 °C was extended to 192 h, there were a small number of equilibrium η phases in the aluminum plates that were completely incoherent with the matrix and destroyed the continuity of the aluminum matrix, easily causing stress concentration. As a result, the fatigue life of alloy plates was shortened to 9.422 × 10^5^ cycles.

## 1. Introduction

The 7005 aluminum alloy has been widely used in rail transit vehicles and the aerospace field as a key structural material due to its high specific strength [1], good weldability [2], and hot-working performance [3]. The structural materials used in rail transit vehicles and aircraft are subjected to cyclic alternating stress for long durations during service; therefore, aluminum alloys must exhibit both high strength and plasticity and good fatigue performance.

The configuration of precipitated particles is one of the key microstructural factors that determines the strength, plasticity, and fatigue properties of aging-hardened aluminum alloys. Liang et al. [4] studied 2024 aluminum plates and found their yield and tensile strength to first increase and then decrease upon extending the aging time from 4 to 36 h at 185 °C. The yield and tensile strengths of the alloy plates peaked when aged for 24 h (388 MPa) and 16 h (476 MPa), respectively, while their elongation decreased monotonically from 22.0% to 9.7%. Zang et al. [5] found that the yield and tensile strength of 2124 aluminum alloy plates increased monotonically, while their elongation decreased monotonically with increasing aging time, when aged at 170, 175, and 180 °C. Wang et al. [6] found the tensile and yield strengths of squeeze-cast Al–4.91Zn–1.99Mg–1.5Cu alloy to increase when the aging was extended from 2 h to 8 h at 135 °C, while the elongation decreased. In conclusion, the results of the present study show that the strength of an aluminum alloy first increases and then decreases with increasing aging time, but the corresponding elongation decreases monotonically [7].

However, there are two prevalent views on the effect of precipitate particles on the fatigue properties of aluminum alloy plates [8]. A few reports have suggested that small GP zones, which are completely coherent with the matrix, contribute to improving the fatigue performance of aluminum alloy plates. Chen et al. [9] found that small GP zones, which were completely coherent with the matrix, were conducive to improving the fatigue crack growth resistance of 7055 aluminum alloy plates based on the reversible slip model of dislocation. Xue et al. [10] studied the fatigue crack growth rate of 2A97 aluminum-lithium alloy sheets after different aging treatments and found that when ΔK was low, the crack growth rates of the alloy plates were significantly different. Transition-phase particles with coarse sizes and irregular edges in T6 tempered (165 °C × 48 h) alloy plate matrix easily led to stress concentration, which sharply decreased fatigue crack growth resistance. Therefore, the crack growth rate of the T6 tempered alloy sheet was the fastest. In contrast, some reports indicate that the large size of semi-coherent metastable phase or equilibrium phase particles, which are completely incoherent with the matrix, improves the fatigue performance of aluminum alloy plates. Suresh et al. [11] showed that the larger equilibrium η-phase particles precipitated in 7075 aluminum alloy plates by overaging treatment were easily oxidized during the fatigue cyclic loading process, leading to the closure effects of induced oxide closure and particle plugging at the crack tip. The fatigue crack growth rate of aged alloy plates was the lowest compared to under-aged and peak-aged plates. Gurbuz et al. [12] also concluded that large precipitated particles that could not be cut through fatigue cracks were conducive to reducing the fatigue crack growth rate of the alloy. Desmukh et al. [13] found that the coarse metastable η′-phase and equilibrium η-phase particles in over-aged 7010 aluminum alloy, which could not be cut through dislocations, promoted a uniform plastic deformation of the matrix, conducive to reducing the fatigue crack growth rate and improving the fatigue strength of the alloy.

In this work, the effect of the configurations of the precipitates on the tensile and fatigue properties of 7005 aluminum alloy plates was systematically studied, elucidating the affecting mechanism of the crystal lattice type, size, and number of precipitates on the strength, plasticity, and fatigue fracture behavior of the alloy plate.

## 2. Materials and Methods

The 3.3 mm-thick hot-rolled 7005 aluminum alloy plate used in the present study was purchased from Northeast Light Alloy Co., Ltd (Harbin, China). The composition of 7005 aluminum alloy plates was as follows (wt.%): 4.67 Zn, 1.04 Mg, 0.13 Cu, 0.28 Mn, 0.17 Zr, 0.17 Cr, 0.132 Fe, 0.029 Si, <0.1 others, and the balance Al. The tensile and fatigue specimens were cut from the alloy plate along the rolling direction, and the loading direction of the tensile and fatigue tests was consistent with the rolling direction. Their corresponding specific dimensions are shown in Figure 1. All specimens were treated with the solution at 470 °C for 90 min in a resistance furnace and then water-quenched. The resting time at room temperature was not more than 10 min, following which they were aged and held in an air furnace at 120 °C for 1, 2, 4, 6, 8, 12, 24, 96, and 192 h, consecutively. The tensile experiments were conducted using a Shimadzu AG-X 250 KN electronic universal testing machine (Shimadzu Corporation, Kyoto, Japan) at room temperature (25 °C) at a strain rate of 1 × 10^−3^ s^−1^ and data acquisition frequency of 100 Hz. The load accuracy was 100 N, and stress and strain were measured using a force transducer and extensometer, respectively. Fatigue experiments were conducted using a QBG-100 high-frequency fatigue testing machine (Sinotest Equipment Co. Ltd., Changchun, China) at room temperature. A fatigue loading stress of 185 MPa, stress ratio (R) of 0, and sinewave as the loading wave were used. Five specimens were tested for both the tensile and fatigue experiments.

The electrical conductivity of the specimens was measured using a SigmaScope SMP10 Fischer (HelmutFischer, Sindelfingen, Germany) conductivity instrument at room temperature (25 °C). Six points on the rolling plane of the alloy plate surface were measured for the conductivity test to obtain the average value. The grain microstructures were anodic coated with a film at a voltage of 25 V, and the composition of the coating solution was Barker reagent (1% HF + 1% HBF_4_ + 24% C_2_H_5_OH + 74% H_2_O). The microstructures of the test samples were determined via optical microscopy (OM) (OLYMPUS GX71, Olympus Corporation, Tokyo, Japan) and transmission electron microscopy (TEM) (JEM-2100F, JEOL Ltd., Tokyo, Japan) at an accelerating voltage of 200 kV, and Image-Pro Plus 6.0 (IPP) was used to determine the number and size of precipitated particles in 7005 aluminum alloy plates aged at 120 °C for 24 h, 96 h, and 192 h. Specimens for TEM analysis were cut from the 7005 aluminum alloy plates after undergoing different aging treatments, thinned to approximately 80 µm, and subsequently electropolished in a twin-jet polishing unit (MTP-1A Magnetic Twin-jet Electropolisher, Shanghai Jiaoda Electromechanical Technology Development Co. Ltd., Shanghai, China). This unit was operated at 25 V and −25 °C using a 25% nitric acid and 75% methanol solution until perforation occurred. The fatigue fracture initiation and crack propagation morphology were investigated via scanning electron microscopy (SEM) (JSM-6510A, JEOL Ltd., Tokyo, Japan).

## 3. Results

### 3.1. Microstructure of 7005 Alloy Plates after Aging at 120 °C

Figure 2 shows the OM images of the 7005 aluminum alloy plates aged at 120 °C for different durations: 2–15 μm of dark gray clumps of massive excess constituent particles were visible with fragmented chain distributions along the rolling direction in all the alloy plates. According to the results of Shan [14] and Ji [15], the 2–15 μm of dark gray clumps of excess constituent particles are presumed to be insoluble in α-Al(FeMn)Si (indicated by the arrow in Figure 2b). No evident differences were found in configuration of the micron-sized insoluble excess constituent particles of α-Al(FeMn)Si between the alloy plates aged at 120 °C for different durations, indicating that aging at 120 °C for less than 192 h had no significant effect on the configuration of the excess constituent particles in the 7005 aluminum alloy plates.

Figure 3 shows the images of the grain in the longitudinal section of the 7005 aluminum alloy plates aged at 120 °C for different durations. The grain size and shape of all these plates are observed to be roughly the same, and they all exhibited elongated fibrous characteristics with their long axes being in the rolling direction. The results indicate that non-recrystallization occurred after water quenching at 470 °C for 90 min, and the subsequent aging treatment had no evident effect on the grain size and shape of the alloy plates.

Figure 4 shows the BF-TEM images of grain boundary precipitates (GBPs) of the 7005 aluminum alloy plates aged at 120 °C for different durations. Precipitates at the grain boundaries of the alloy plates are observed for aging durations greater than 1 h at 120 °C. When the aging time does not exceed 12 h, the precipitates are small (1–7 nm) and are distributed intermittently. As the aging time increases from 1 to 12 h, the size of precipitates gradually increases (Figure 4a–d). When the aging holding time is extended to 24 h, the grain boundary precipitates continue to grow up to 28 nm, with small gaps between the particles, exhibiting a slightly continuous distribution characteristic. A precipitate free zone (PFZ) with a width of about 50 nm is observed near the grain boundaries (Figure 4e). When the time of aging at 120 °C is extended to 96 h, the size of the grain boundary precipitates continues to grow up to 31 nm and exhibit a relatively evident intermittent distribution characteristic. The PFZ width remains unchanged at about 50 nm (Figure 4f). When aging time is extended to 192 h, the size of the grain boundary precipitates and the width of PFZ remain unchanged (Figure 4g).

Figure 5 shows the BF-TEM and high-resolution transmission electron microscopy (HREM) images of the matrix precipitates (MPts) of 7005 aluminum alloy plates aged at 120 °C for different durations. It is observed that after aging at 120 °C, there are some spherical Al_3_Zr dispersed-phase particles sized 12–25 nm in the alloy plate matrix (as indicated by the arrows in Figure 5a,c,e,g) [16], and the aging time at 120 °C has no evident effect on the configuration of their dispersal. After aging at 120 °C for 1 and 4 h, the precipitates in the alloy plates are GP zones completely in common with the matrix, the number and size of the precipitates gradually increase with aging time, and the distribution is more uniform (Figure 5a–d). When the aging time reaches 8 h, in addition to the GP zones, an extremely small metastable η′ phase is observed in the alloy plate matrix (Figure 5e,f). When the aging time is extended to 12 h, the GP zones in the alloy plate matrix gradually change to metastable η′-phase particles, and the volume fraction of the GP zones decreases. However, the size of the metastable η′-phase particles does not change significantly, although the number of metastable particles gradually increase (Figure 5g,h). This is consistent with the results observed by Shao et al. [17] in Al–Zn–Mg aluminum alloy. When the aging time is extended to 24 h, the number of GP zones in the alloy plate matrix decrease further, while the metastable η′-phase particles gradually grow to 3.7 nm (Figure 5i,j). When the aging time is extended to 96 h at 120 °C, almost all of the matrix precipitates of the alloy plates are metastable η′-phase particles with size of 5.1 nm, and the number of precipitates increases significantly (Figure 5k,l). When the aging time is extended to 192 h at 120 °C, a small number of equilibrium η-phase particles start to appear in the alloy plate matrix, and the average size of both the η′ and η particles is about 6.9 nm, but the number of precipitates remain the same (Figure 5m,n).

### 3.2. Conductivity and Mechanical Properties of 7005 Alloy Plates after Aging at 120 °C

The electrical conductivity curves of the 7005 aluminum alloy plates aged at 120 °C for different durations are shown in Figure 6, wherein they exhibit a trend of gradually increasing with increasing aging time. The conductivity of the alloy plates increased slowly when the aging time was less than 6 h. However, the conductivity of the alloy plates remained unchanged when the aging time increased from 6 to 12 h, i.e., 37.2% IACS. When the aging time was more than 12 h, the conductivity of the alloy plates increased continuously, and when the aging time was extended to 192 h, the conductivity of the alloy plates reached its maximum value, i.e., 40.9% IACS. A comprehensive analysis showed that the exsolution precipitation of the solute Mg and Zn atoms in the alloy plate matrix increased with increasing aging time [18]. The lattice distortion and the scattering effect on electrons are both reduced, resulting in the electrons flowing more easily and leading to an increase in electrical conductivity. This was consistent with the results of Ozer et al. [19] in aging precipitation of AA7075 aluminum alloy sheet.

Based on the electrical conductivity curves, 7005 aluminum alloy plates treated under the same aging conditions were chosen to test the room-temperature tensile properties, and the corresponding curves are presented in Figure 7. It can be seen that the yield and tensile strengths of the alloy plates increase monotonically with increasing aging time, while the elongation of the alloy plates show a trend of first increasing and then decreasing continuously. The yield and tensile strengths of the alloy plates increase from 168 and 255 MPa to 171 and 263 MPa, respectively, and the elongation of the plates increased from 19% to a peak of 23% upon increasing the aging time at 120 °C from 1 h to 4 h. Upon increasing the aging time continuously to 192 h, the yield and tensile strengths of the alloy plates reach a maximum of 340 MPa and 382 MPa, respectively, and the corresponding elongation reduced to a minimum of 14%. The 7005 alloy plates exhibit an anomalous phenomena in that the strength and elongation simultaneously increase when the aging time is less than 4 h. This was not consistent with other research results obtained by most current scholars [20,21] that showed aluminum alloy elongation decreasing while its strength increased with increased aging time.

It is worth noting that the increments in the electrical conductivity, yield strength, tensile strength, and elongation of the alloy plates are only 0.4% IACS, 2 MPa, 4 MPa, and −1%, respectively, when the aging time is extended from 96 h to 192 h. This shows that upon aging at 120 °C for 192 h, the solution atoms in the alloy plate matrix precipitate almost completely, and the alloy plates reach the peak aging state. This also shows that 7005 aluminum alloy exhibits good aging stability at 120 °C, consistent with the results obtained by Li [22].

### 3.3. Fatigue Life of 7005 Alloy Plates after Aging at 120 °C

The fatigue-life data of 7005 aluminum alloy plates aged at 120 °C for different durations are presented in Figure 8 (fatigue stress level, 185 MPa; stress ratio (R) = 0). It is worth noting that the yield strengths of the alloy plates aged at 120 °C for 1 and 4 h are 168 and 174 MPa, respectively, and those of the alloy plates aged at 120 °C for more than 8 h are all greater than 193 MPa; furthermore, the fatigue stress level = 185 MPa is higher than that at 174 MPa but less than that at 193 MPa. Therefore, under this condition, the fatigue deformation behavior of the alloy plates aged for 1 and 4 h at 120 °C is different from that of the alloy plates aged more than 8 h. It is observed in Figure 8 that the fatigue life of plates aged at 120 °C for 8 h is the shortest: only 1.20 × 10^5^ cycles. Moreover, the fatigue life of plates aged at 120 °C for 1 and 4 h are 3.297 × 10^5^ and 3.544 × 10^5^ cycles, respectively, longer than those of the alloy plates aged at 120 °C for 8 h. When the aging time exceeds 8 h, the fatigue life of the alloy plates shows a trend of extending significantly first and then shortening slightly. The fatigue life of the alloy plates aged for 96 h is the longest, i.e., 1.272 × 10^6^ cycles, 10.6 times the fatigue life of the alloy plates aged at 120 °C for 8 h. The fatigue life of the alloy plates aged for 192 h slightly decreases to 9.422 × 10^5^ cycles. It is 25.9% shorter than that of the alloy plates aged for 96 h, but 65.5% longer than that of the alloy plates aged for 24 h (5.693 × 10^5^ cycles), i.e., 7.85 times the fatigue life of the alloy plates aged at 120 °C for 8 h.

### 3.4. Fatigue Fracture Morphologies of 7005 Alloy plates Aged at 120 °C for Different Durations

Figure 9 shows the SEM micrographs of the fatigue fracture initiation of 7005 aluminum alloy plates aged at 120 °C for different durations at a fatigue stress level of 185 MPa and stress ratio (R) = 0. The fatigue cracks of all the aged plates at different durations initiated in the coarse excess constituent particles at the surface or near the surface of the specimens, and the fatigue cracks extended radially from the fatigue crack source to the inner part of the plates. Under cyclic loading, the dislocation slips to the interface of the coarse excess constituent particles and piles up, resulting in stress concentration, which leads to the separation of the interface between the excess constituent particles and matrix to form holes. This is in accordance with the results obtained by Luo [23]. The characteristics of the fatigue fracture morphology in the fatigue crack initiation region of the alloy plates aged at 120 °C for 1, 4, 12, and 24 h are almost identical: the fracture surface has less undulation, and the crystallographic planes are relatively small (Figure 9a,b,d,e). The fluctuation degree of the fatigue-cracked surface in the fatigue crack initiation region is relatively large for the alloy plates aged at 120 °C for 8, 96, and 192 h, and the corresponding crystallographic plane features are more significant (Figure 9c,f,g). In conclusion, the precipitation phase particle configuration had no significant effect on the location of fatigue crack initiation; however, there were some differences in fatigue crack initiation and extension behavior of the alloy plates with different precipitation phase particle configurations.

Figure 10 shows the SEM micrographs of fatigue fracture crack propagation surfaces of 7005 aluminum alloy plates aged at 120 °C for different durations at a fatigue stress level of 185 MPa and stress ratio (R) = 0. Simultaneously, parallel fatigue striations are observed in the high-power SEM micrographs. The surface morphology of the fracture crack propagation zone of the alloy plates is rough after aging at 120 °C for 1 and 4 h. The surface, which is mainly composed of river-like stripes and tearing edges, exhibits evident river pattern characteristics, whereas fatigue striations and a small number of long, furrow-shaped secondary cracks are observed on the fracture surface (Figure 10a,c). The fatigue striation space is 0.82 and 0.85 μm (Figure 10b,d), consistent with the surface morphology of the fracture crack propagation zone observed by our research group studying the effect of pre-tensile deformation on fatigue fracture of under-aged 7N01 aluminum alloy plates [24]. This suggests that the fatigue crack propagation direction in the alloy plate is deflected numerous times, and the deflection angle is small. The fracture crack propagation zone surface of the alloy plates exhibits a large degree of undulation and evident crystallographic plane characteristics after aging at 120 °C for 8 and 12 h. A large number of tearing edges and fatigue steps are observed near these steps (Figure 10e,g). In addition, it is observed that the fatigue striation space of the alloy plates aged at 120 °C for 8 h is the widest, i.e., 1.59 μm (Figure 10f), indicating that the fatigue crack growth rate of the alloy plates aged at 120 °C for 8 h is the highest. The surface of the fatigue fracture crack propagation zone of the alloy plates aged at 120 °C for 24 and 96 h is relatively smooth, the fracture surface has less undulation, and the area of the crystallographic planes is significantly reduced. A small number of secondary cracks are observed in the fatigue crack propagation paths (Figure 10i,k). In addition, the fatigue striation space of the alloy plate aged at 120 °C for 96 h is the narrowest, i.e., 0.28 μm (Figure 10l), indicating that the fatigue crack growth rate of the alloy plates aged at 120 °C for 96 h is the smallest. The fracture crack propagation zone surface of the alloy plates aged at 120 °C for 192 h is relatively rough, has a large degree of undulation (Figure 10m), and the corresponding fatigue striation space is 0.31 μm (Figure 10n).

## 4. Discussion

The results in Figure 2, Figure 3, and Figure 5 show that extending the aging time at 120 °C from 1 h to 192 h does not have a significant effect on the configuration of the micron-sized α-Al(FeMn)Si excess constituent particles, the sub-micron Al_3_Zr dispersoid particles, or the grain morphology and size in 7005 aluminum alloy plates. Since the micron-sized α-Al(FeMn)Si crystalline particles in the alloy plate were formed during the solidification process of the ingot and then broken into fragmented chain distributions along the rolling direction by hot-rolling, the sub-micron Al_3_Zr dispersoid particles precipitated during the homogenization process of the ingot and the shape, size, and quantity of these dispersoid particles were not significantly affected by hot-rolling. Solution treatment at 470 °C and aging treatment at 120 °C did not change the configuration of the excess constituents and dispersoid particles. Meanwhile, the grains of the alloy matrix were still fibrous after exposure to solution at 470 °C for 90 min because the sub-micron-sized dispersoid particles in the alloy plates pinned the grain boundaries and inhibited them from arching out; i.e., recrystallization nucleation was inhibited. The aging treatment at 120 °C had almost no effect on the configuration of the excess constituent particles, dispersoid particles, or grains of the aluminum alloy plates. This is in accordance with the results obtained by Kazakova [25].

From the combination of alloy plate TEM images (Figure 4 and Figure 5) and electrical conductivity change curves (Figure 6) when the alloy plates had been aged at 120 °C, the solute atoms of Mg and Zn had been continuously exsolved and precipitated from the aluminum matrix in the form of GP zones and η′- and η-phase particles. Almost all the precipitates at the grain boundary are equilibrium η-phase particles, and the size of the precipitates at the grain boundary is evidently larger than that of the matrix. A PFZ with a width of 50 nm is observed near the grain boundary when the aging time exceeds 24 h, but there is no significant effect on the PFZ width when aging time extends to 192 h. When the aging time is extended from 1 to 192 h, the equilibrium η-phase precipitates at the grain boundary become coarser and disperse gradually. The matrix precipitates undergo the following transformation: GP zone nucleation stage (GP zones show no evident change in size but quantity increases) → GP zone growth stage (the size of GP zones increases but the quantity remains approximately the same) → GP zone to metastable η′-phase particle transformation stage (a large quantity of GP zones + a small quantity of η′) → quantity of metastable η′-phase particles increases (a small quantity of GP zones + a large quantity of η′) → all metastable η′-phase particles → metastable η′-phase particle to equilibrium η-phase particle transformation (a large quantity of η′ + a small quantity of η). When the time of aging at 120 °C is less than 4 h, the aging precipitate of the alloy plate matrix is in the nucleation and growth stage of the GP zones, and a large number of GP zones are constantly being formed in the alloy plates, and they are completely coherent with the matrix. When the aging time is extended from 1 to 4 h, the quantity and size of the GP zones increase, and the distribution is more uniform. This is consistent with the results reported by Sha [26]. When the aging time exceeds 4 h, metastable η′-phase particles begin to appear in the alloy matrix, and some GP zones in the alloy matrix begin to transform into metastable η′-phase particles. When the aging time is extended from 4 to 96 h, the quantity of the GP zones decreases, the quantity of metastable η′-phase particles gradually increases, and the η′ particles grow constantly as the Mg and Zn solute atoms in the solid solution continue to dissolve in the matrix. When the aging time is extended to 96 h, almost all the precipitates in the grains of the alloy plates are metastable η′-phase particles, consistent with the results observed by Li et al. [27] in Al–5.4Zn–1.2Mg-0.16Zr aluminum alloy. When the aging time extends from 96 to 192 h, solute atoms in the alloy matrix continue to dissolve and precipitate, metastable η′ particles continue to grow, and a small quantity of equilibrium η particles begin to appear in the grains of the alloy plates. This is consistent with the research results of Engdahl et al. [28]. In summary, it was seen that the crystal structure, size, and number of precipitation-phase particles in the 7005 aluminum alloy plates changed during the aging process at 120 °C after solid solution exposure and then water-quenching, but the evolution of precipitates was very slow, indicating that the quenched 7005 aluminum alloy exhibited good aging stability at 120 °C.

The evolution of precipitates formed by the precipitation of Zn and Mg atoms in the solution from the matrix was the fundamental reason that the strength of the alloy plates increased monotonously; the elongation of the alloy plates first increased and then decreased as the aging time increased. When the aging time was less than 4 h, a large number of GP zones that were completely coherent with the matrix precipitated in the grains of the alloy plates, and as the aging time was prolonged, the size, number, and distribution of the GP zones increased (Figure 5a–d). Owing to the small size of the GP zones and their complete coherence with the aluminum matrix, they had a strong hindering effect on dislocation movement. During tensile deformation, almost all the moving dislocations passed through the GP zones in the cutting mode. The greater the number, size, and distribution of GP zones, the stronger the hindrance to dislocation movement. Therefore, the yield and tensile strengths of the alloy plates increased slightly from 168 MPa and 255 MPa, respectively, after aging for 1 h, to 174 MPa and 264 MPa, respectively, after aging for 4 h. It can be noted that when the aging time was less than 4 h, there was no evident PFZ near the grain boundary, a large number of solute atoms were still dissolved in the matrix near the grain boundary, and small η-phase particles precipitated on the grain boundary. GP zones (clusters of solute atoms) with high density and uniform distribution, PFZ of supersaturated Mg and Zn solute atoms from the grain boundary to intragrain, and small η-phase particles at the grain boundary caused the strength of the alloy plates from the grain boundary to the ingrain to be uniform, which was beneficial to the overall coordinated deformation of the ingrains, grain boundaries, and grains, and the corresponding dislocation slip was uniform. Moreover, with the increase in the number, size, and distribution of GP zones, the intensity difference between intragranular and grain boundaries decreased gradually, the deformation coordination ability within and between the grains of the alloy plates significantly improved, the dislocation slip deformation within and between grains was more uniform, and the more plastic deformation the alloy plates could bear before fracture. Therefore, the elongation of alloy plates increased from 19% after being aged for 1 h to 23% after being aged for 4 h. The increase in the number, size, and uniform distribution of GP zones completely coherent with the matrix was the fundamental reason for the increase in the strength and elongation of the alloy plates with an aging time from 1h to 4 h at 120 °C [29].

When the aging time exceeds 4 h, the large-sized GP zones begin to transform into semicoherent metastable η′-phase particles. While the dissolved Zn and Mg atoms in the matrix of the alloy plates further dissolve and precipitate, the number of intragranular GP zones decreases, and the number of metastable η′-phase particles gradually increases (Figure 5e–j). When the aging time is extended to 96 h, almost all the GP zones are transformed into metastable η′-phase particles (Figure 5k,l). When the aging time is extended to 192 h, a small number of equilibrium η-phase particles having a size of 12 nm begin to appear in the matrix of the alloy plates (Figure 5m,n). Research by Kovács [30] on theoretical calculations of shear stress for dislocation cutting and bypassing precipitated particles for an Al–Zn–Mg alloy showed that when the size of the precipitated phase particles was less than 6 nm, the dislocation passed through the precipitated particles by cutting, while when the size was more than 6 nm, the dislocation generally passed through the precipitated particles by passing. The dislocations mostly bypassed the metastable η′-phase particles and equilibrium η-phase particles because the metastable η′-phase particles were large and semi-coherent with the matrix interface, whereas the equilibrium η-phase particles were larger, and their interface was completely incoherent with the aluminum matrix. As aging time was extended, the size and quantity of metastable η′-phase particles increased, and they began to transform into equilibrium η-phase particles, leading to a significant increase in the resistance of the dislocation slip during plastic deformation; thus, the strength of the alloy plates continued to increase. However, the degree of incoherence between the metastable η-phase particles and equilibrium η-phase particles and the interface of the aluminum matrix gradually increased. The hard and brittle particles that were incoherent with the matrix destroyed the continuity of the matrix of the alloy plates, which was unconducive to uniform deformation. The larger their number and size, the more unfavorable was the overall coordinated deformation of the matrix. Therefore, when metastable η′-phase particles began to appear in the matrix of the alloy plates, the fracture elongation of the alloy plates gradually decreased with increasing aging time. The yield and tensile strengths of the alloy plates reached their highest, i.e., 340 and 382 MPa, respectively, when aged at 120 °C for 192 h, but their fracture elongation was the lowest, i.e., only 14%.

The yield strengths of the alloy plates aged at 120 °C for 1 and 4 h were 168 and 171 MPa, respectively. When the maximum fatigue stress level was 185 MPa, a certain degree of plastic deformation inevitably occurred in the alloy plates during the first several loading cycles under fatigue cyclic loading, and a certain number of dislocations occurred in the matrix of the alloy plate. The aluminum matrix exhibited a certain degree of work hardening. Xu et al. [31] believed that the mismatch in strength, stiffness, and plasticity between the second-phase particles and matrix in metal materials was the fundamental reason for the initiation of fatigue cracks at the interface between the second-phase particles and matrix during fatigue cyclic loading. Only some GP zones with extremely small precipitates in the aluminum matrix of the alloy plates (Figure 5a–d) after aging at 120 °C for 1 and 4 h had less strengthening effect on the aluminum matrix. The strength and hardness of the matrix plates were very low compared with those of the large-sized excess constituent particles. A certain degree of plastic deformation occurred in the alloy plate, and the strength and hardness of the aluminum matrix improved after a large number of dislocations occurred in the aluminum matrix when the maximum stress of fatigue cyclic loading surpassed its yield limit. To some extent, the mismatch between the large excess constituent particles and matrix was weakened, and the overall load-bearing coordination of the aluminum plate improved, thereby improving the fatigue performance of the alloy plate to a certain extent. Therefore, the fatigue lives of the plates aged at 120 °C for 1 and 4 h were approximately 3.297 × 10^5^ and 3.544 × 10^5^ cycles, respectively, for the maximum fatigue stress level of 185 MPa and stress ratio (R) = 0. The fatigue life of the alloy plates was longer compared with that of the alloy plate aged at 120 °C for 8 h (1.20 × 10^5^ cycles), consistent with the conclusion of the research group’s previous study on the influence of pre-stretching on the fatigue fracture of under-aged 7N01 aluminum alloy plate [24].

The matrix precipitates in the alloy plates are GP zones, which are completely coherent with the matrix, and a small amount of extremely small metastable η′-phase particles after aging at 120 °C for 8 h (Figure 5e,f). The excess constituent particles and α-Al matrix intensity difference was relatively large, and alloy plate exhibited poor coordination overall due to the increase in the aged alloy plate yield strength to 193 MPa (greater than the maximum stress of 185 MPa for fatigue loading). Consequently, fatigue cracks tended to initiate in large excess constituent particles at the surface or near the surface of the specimens (Figure 10e). In contrast, if alloy plate strength is relatively low and the resistance to fatigue crack growth is small, fatigue crack will spread easily. Although the degree of surface undulation of the fatigue crack propagation zone in the fatigue fracture of the alloy plate is large, the distance is large between the fatigue striations in the fracture, i.e., 1.59 μm (Figure 10e,f). The resistance of the aluminum matrix to fatigue crack initiation and propagation was very small for the alloy plate aged at 120 °C for 8 h at the maximum fatigue stress level of 185 MPa and stress ratio (R) = 0. This was the main reason for the shortest fatigue life of the alloy plate, i.e., 1.20 × 10^5^ cycles.

As the aging time continued to increase, the number of GP zones in the alloy matrix decreased, the number of metastable η′ particles increased, and the yield strength of the alloy plates increased further. The yield strength of alloy plates increased from 216 MPa to 338 MPa when the aging time was extended from 12 to 96 h at 120 °C. The strength difference between the age-strengthened α-Al matrix and large excess constituent particles gradually decreased, overall load-bearing coordination of the alloy plate gradually increased, and the resistance of the aluminum matrix to fatigue crack initiation gradually increased; thus, the fatigue life of the alloy plates corresponding to the fatigue crack initiation stage gradually extended. In contrast, the age-strengthened α-Al matrix made fatigue crack propagation more difficult, its resistance to fatigue crack propagation increased, and the fatigue striation spaces in the fatigue fracture of alloy plates decreased gradually from 1.07 μm (120 °C aging for 12 h, Figure 10h) to 0.28 μm (120 °C aging for 96 h, Figure 10l). A comprehensive analysis showed that the increased number of metastable η′-phase particles gradually increased the strengthening effect on the alloy plate matrix. The mismatch between the matrix and large excess constituent particles decreased, the overall load-bearing coordination ability of the alloy plate improved, and the resistance to fatigue crack initiation improved. The fatigue life of the alloy plates corresponding to the fatigue crack initiation stage also gradually extended. However, the resistance of the reinforced aluminum matrix to fatigue crack propagation was enhanced. This was beneficial for extending the fatigue life of the alloy plates corresponding to the fatigue crack propagation stage. The superposition of the two caused the fatigue life of the alloy plates aged at 120 °C for 12, 24, and 96 h to gradually extend to 1.459 × 10^5^, 5.693 × 10^5^, and 1.272 × 10^6^ cycles, respectively.

The precipitated phase in the grains of the alloy plates began to transform from metastable η′-phase particles to equilibrium η-phase particles when the aging time further extended to 192 h at 120 °C (Figure 5m,n), while the yield strength of the alloy plate increased by only 2 MPa. This indicated that the strength difference between the α-Al matrix and excess constituent particles did not change significantly upon extending the aging time from 96 to 192 h. However, the equilibrium η-phase particles, which were completely incoherent with the matrix and, critically, destroyed the continuity of the aluminum matrix, easily caused stress concentration, which was unconducive to the overall load-bearing coordination of the alloy plates, resulting in a slight decrease in the resistance of the alloy matrix to fatigue crack initiation and propagation. Therefore, the resistance of the alloy plates to fatigue crack initiation decreased, and the fatigue life of the alloy plates corresponding to fatigue crack initiation slightly shortened, but it was still higher than the life of the alloy plate corresponding to fatigue crack initiation when aged at 120 °C for 24 h. The distance between the fatigue striations in the fracture crack propagation zone of the alloy plates was 0.31 μm, slightly larger than the 0.28 μm in the fracture crack propagation zone of the alloy plates aged for 96 h. However, it was much smaller than that of the fracture crack propagation zone in the fatigue fracture of the alloy plates aged for 24 h (0.70 μm), In other words, the resistance of fatigue crack propagation was greater for the alloy plates aged at 120 °C for 192 h compared with those aged for 24 h. In conclusion, the continuity of the aluminum matrix was destroyed due to the presence of equilibrium η-phase particles, which were completely incompatible with the matrix after aging at 120 °C for 192 h. This easily caused stress concentration and reduced the resistance of the plate matrix to both fatigue crack initiation and propagation. As a result, the fatigue life of the alloy plates was shortened by 25.9% to 9.422 × 10^5^ cycles when the aging time at 120 °C was extended from 96 h to 192 h. Meanwhile, the fatigue life of the alloy plates aged at 120 °C for 192 h was 65.5% longer than that of the alloy plates aged for 24 h (5.693 × 10^5^ cycles).

A schematic diagram of the effects of precipitation particle configuration on fatigue crack initiation and propagation in 7005 aluminum alloy plates was created based on the above discussion and analysis, as shown in Figure 11. When Zn and Mg mainly exist in the matrix of the 7005 aluminum alloy plates as solid solute atoms or extremely small GP zones, their strengthening effect on the matrix is small, and the yield strength of the plates is lower than the maximum fatigue stress level of 185 MPa. A certain degree of plastic deformation and a number of dislocations occur in the first few loading cycles when the alloy plates are subjected to cyclic fatigue loading over their yield strength, resulting in the dislocation hardening effect for the alloy matrix. This weakens the mismatch between the large excess constituent particles and matrix to a certain extent, improves the overall load-bearing coordination of the aluminum plate, and delays fatigue crack initiation and propagation in the alloy plates. Fatigue cracks tend to initiate at the interface of micron-sized constituent particles near the surface of the plate. Aluminum matrix with low hardness and good plasticity has little resistance to fatigue crack propagation, the deflection secondary cracks are often induced along the path of main crack propagation, releasing the energy of the main crack to a certain extent, and the alloy plates exhibit good fatigue properties (Figure 11a,b).

Plastic deformation does not occur in the 7005 aluminum alloy plates during fatigue cyclic loading after aging for more than 8 h because the yield strength of the plates is higher than the maximum fatigue stress level of 185 MPa. The strengthening effects of Zn and Mg on the matrix are weak when they mainly exist in the GP zones and a as small amount of metastable η′-phase particles in the alloy plate matrix. The strength difference between the α-Al matrix and excess constituent particles is large, overall load-bearing coordination of the alloy plates is poor, and resistance to fatigue crack initiation and propagation is small. Since aluminum matrix with low hardness and good plasticity has the least resistance to fatigue crack propagation, the fatigue crack propagates along the direction of maximum shear stress: no evident multiple deflections occur during crack propagation, secondary cracks do not easily appear, and the alloy plates exhibit poor fatigue properties (Figure 11c,d). Zn and Mg mainly exist either in the matrix of the alloy plates as metastable η′-phase particles and a small number of GP zones, or entirely in the form of metastable η′-phase particles, and their strengthening effect on the matrix gradually increases as the GP zones transform into metastable η′-phase particles. The strength difference between the α-Al matrix and excess constituent particles decreases gradually, and the overall load-bearing coordination of the aluminum plate increases. Therefore, the resistance of the aluminum alloy plate matrix to fatigue crack initiation and propagation increases correspondingly, and it is difficult to initiate fatigue cracks. In addition, the direction of the main fatigue crack deflects several times in the process of propagation, and many secondary cracks accompany the process of propagation, effectively releasing the stress concentration at the tip of the main crack and causing the main fatigue crack to propagate slowly (Figure 11e,f). Therefore, the fatigue life of the alloy plate is significantly prolonged. The equilibrium η-phase particles are completely incoherent with the matrix, destroying the continuity of the aluminum matrix and easily causing stress concentration when a small amount of metastable η′-phase particles transform into equilibrium η-phase particles, which is unconducive to the overall load-bearing coordination of the alloy plates. As a result, the resistance of the alloy plate matrix to fatigue crack initiation and propagation decreases slightly, the direction of the main fatigue crack gets deflected many times in the process of propagation (Figure 11g), and fatigue life of the alloy plate is slightly reduced. However, it remains higher than the resistance to fatigue crack initiation and propagation in the alloy plates aged for 24 h.

## 5. Conclusions

(1)Both the strength and elongation of the alloy plates increased with extended aging time at 120 °C up to 4 h. Since both the number and size of the GP zones, which were completely coherent with the matrix, increased, the distribution of the GP zones was more uniform in the 7005 aluminum plates.(2)The fatigue life of the alloy plates shortened slightly at first, then significantly prolonged, and then shortened again with the aging time extending from 1 h to 192 h at a fatigue stress level = 185 MPa and R = 0. The fatigue life of the alloy plates aged at 120 °C for 8 h was the shortest, i.e., only 1.20 × 10^5^ cycles, since the precipitates in the alloy plate matrix were mainly GP zones and a small amount of metastable η′ particles. During aging at 120 °C for 96 h, almost all the precipitates in the alloy plate matrix were metastable η′-phase particles, which had the optimal aging strengthening effect on the alloy matrix, and the degree of mismatch between the α-Al matrix and second-phase particles was the smallest; the fatigue crack initiation and propagation resistances were the largest, leading to the best fatigue performance of alloy plates, and the corresponding fatigue life of the aluminum alloy plates was up to 1.272 × 10^6^ cycles. When the aging time was extended to 192 h, there was a small amount of equilibrium η-phase particles in the aluminum plates that were completely incoherent with the matrix, destroying the continuity of the aluminum matrix, easily causing stress concentration, and reducing the overall load-bearing coordination of the alloy plates. As a result, the fatigue life of alloy plates was shortened to 9.422 × 10^5^ cycles, but it was still 65.5% longer than that of the plates aged at 120 °C for 24 h, i.e., 5.693 × 10^5^ cycles.

## Figures and Tables

**Figure 1 materials-15-05951-f001:**
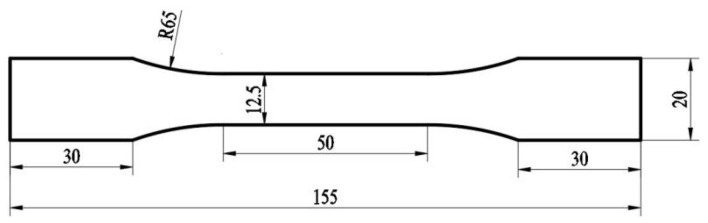
Specimen size of tensile and fatigue test (unit: mm).

**Figure 2 materials-15-05951-f002:**
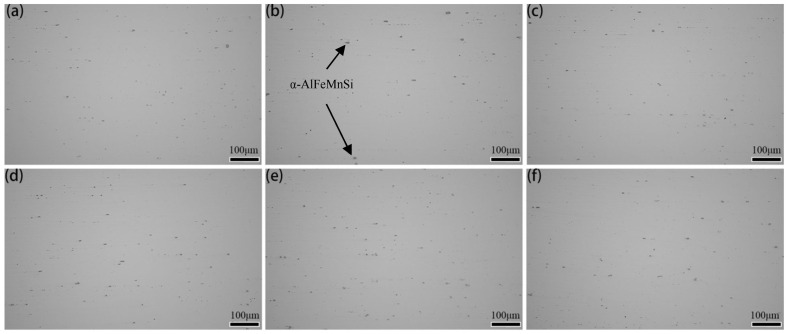
Optical microscopy (OM) images of 7005 aluminum alloy plates aged at 120 °C for different durations (along the longitudinal section, non-etched): (**a**) 1 h, (**b**) 4 h, (**c**) 12 h, (**d**) 24 h, (**e**) 96 h, and (**f**) 192 h.

**Figure 3 materials-15-05951-f003:**
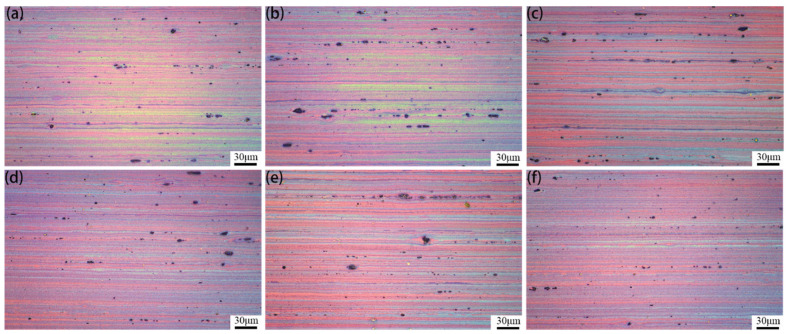
Grains of 7005 aluminum alloy plates aged at 120 °C for different durations (along the longitudinal section, after anodic coated by 1% HF + 1% HBF_4_ + 24% C_2_H_5_OH + 74% H_2_O): (**a**) 1 h, (**b**) 4 h, (**c**) 12 h, (**d**) 24 h, (**e**) 96 h, and (**f**) 192 h.

**Figure 4 materials-15-05951-f004:**
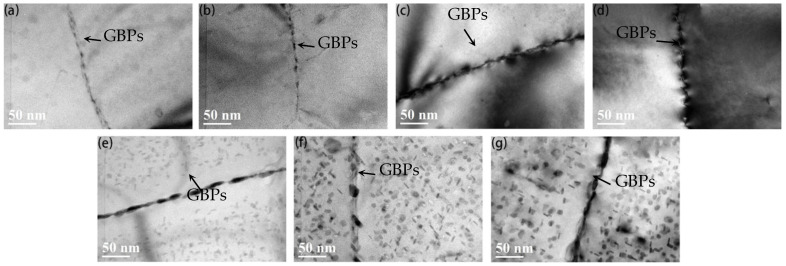
Bright-field transmission electron microscopy (BF-TEM) images of 7005 aluminum alloy plate grain boundary precipitates aged at 120 °C for different durations: (**a**) 1 h, (**b**) 4 h, (**c**)8 h, (**d**) 12 h, (**e**) 24 h, (**f**) 96 h, and (**g**) 192 h.

**Figure 5 materials-15-05951-f005:**
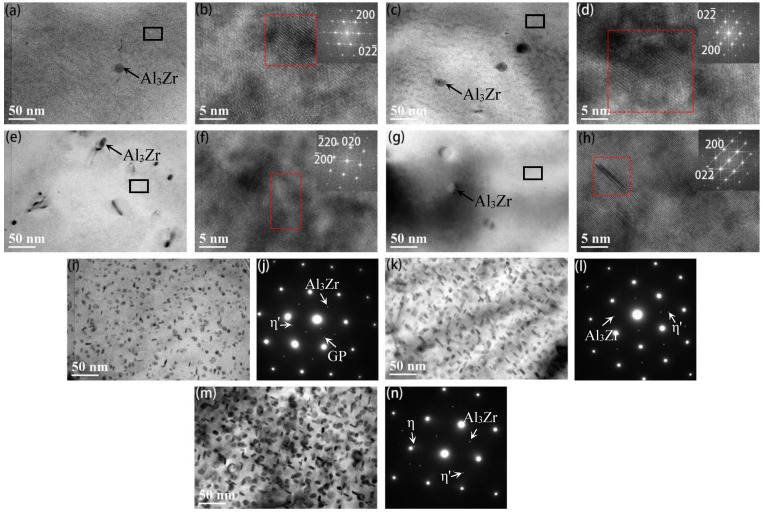
BF-TEM and HREM images of 7005 aluminum alloy plate matrix precipitates along the [110]_Al_ (**a**–**d**,**g**–**n**) and [100]_Al_ (**e**,**f**) orientation aged at 120 °C for different durations: (**a**,**b**) 1 h, (**c**,**d**) 4 h, (**e**,**f**) 8 h, (**g**,**h**) 12 h, (**i**,**j**) 24 h, (**k**,**l**) 96 h, and (**m**,**n**) 192 h. Note: (**b**,**d**,**f**,**h**) are the extracted HREM images of the black boxes in (**a**,**c**,**e**,**g**), respectively, and the red boxes in (**b**,**d**,**f**,**h**) are the Fast Fourier Transformation (FFT) pattern areas in the upper right corner.

**Figure 6 materials-15-05951-f006:**
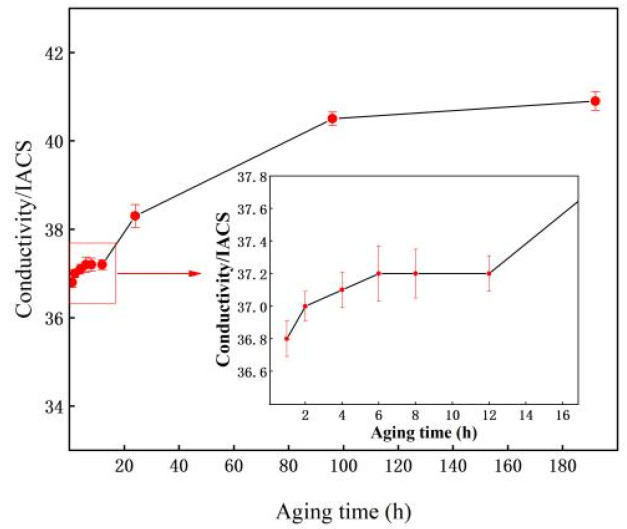
Electrical conductivity curves of 7005 aluminum alloy plates aged at 120 °C for different durations.

**Figure 7 materials-15-05951-f007:**
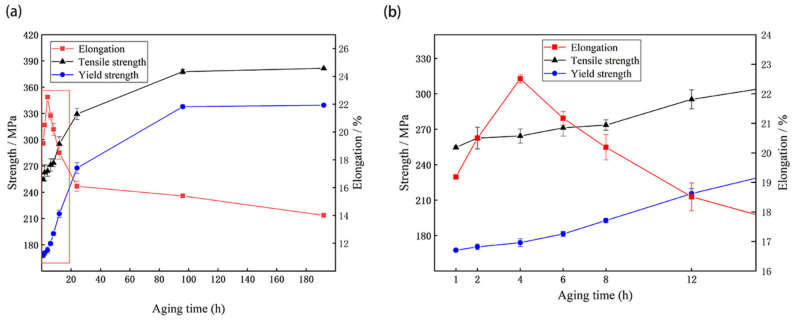
Mechanical performance of 7005 aluminum alloy plates aged at 120 °C for different durations. Note: (**b**) is a partial enlargement of the red box in (**a**).

**Figure 8 materials-15-05951-f008:**
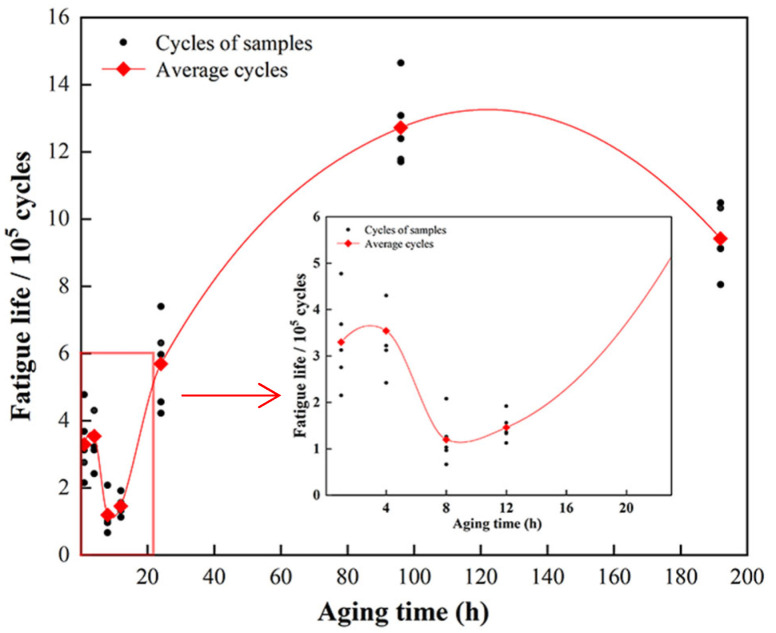
Fatigue life of 7005 aluminum alloy plates aged at 120 °C for different durations (stress level, 185 MPa; stress ratio (R) = 0).

**Figure 9 materials-15-05951-f009:**
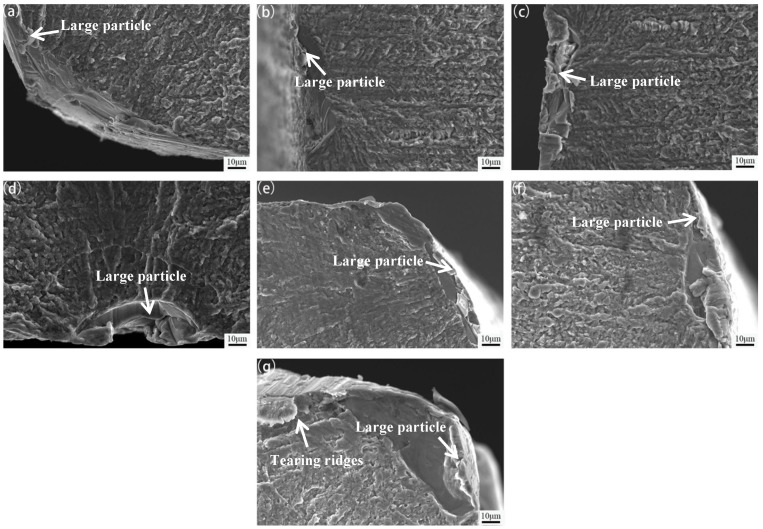
Scanning electron microscopy (SEM) images of fatigue fracture initiation in 7005 aluminum alloy plates aged at 120 °C for different durations: (**a**) 1 h, (**b**) 4 h, (**c**) 8 h, (**d**) 12 h, (**e**) 24 h, (**f**) 96 h, and (**g**) 192 h.

**Figure 10 materials-15-05951-f010:**
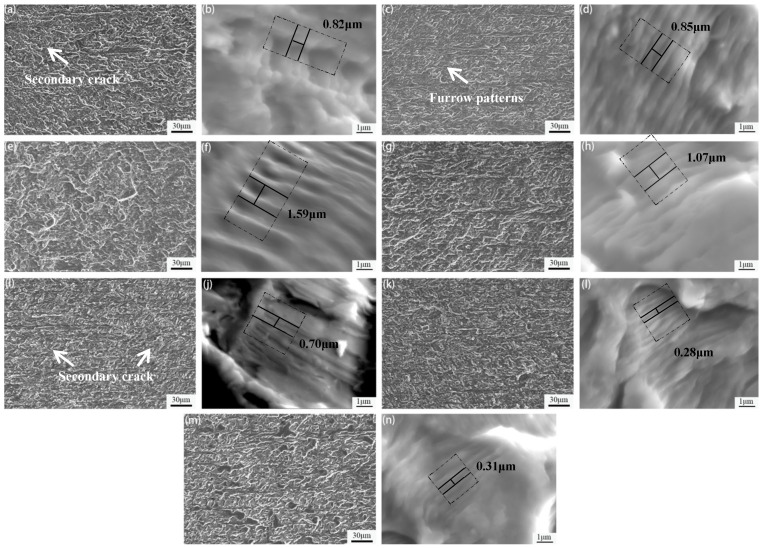
SEM images of fatigue fracture crack propagation in 7005 aluminum alloy plates aged at 120 °C for different durations: (**a**,**b**) 1 h, (**c**,**d**) 4 h, (**e**,**f**) 8 h, (**g**,**h**) 12 h, (**i**,**j**) 24 h, (**k**,**l**) 96 h, and (**m**,**n**) 192 h.

**Figure 11 materials-15-05951-f011:**
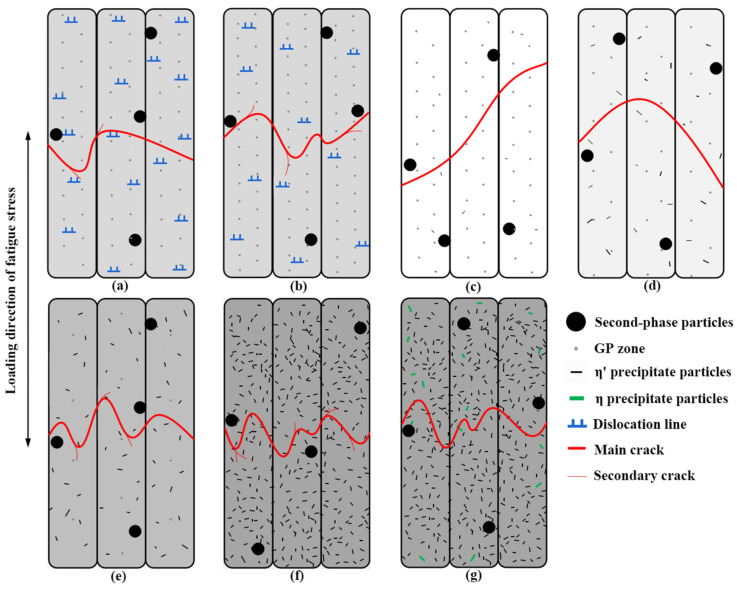
Schematic drawing of fatigue crack initiation and propagation of 7005 aluminum alloy plates with different precipitate configurations: (**a**) small number of GP zones, (**b**) highest volume fraction of GP zones, (**c**–**e**) GP zones + η′, GP zones gradually decrease and η′ gradually increases, (**f**) all η′, and (**g**) large amount of η′ + small amount of η. Note: The matrix color represents the degree of matrix strengthening, with a darker color indicating higher matrix strength.

## Data Availability

The data presented in this study are available on request from the corresponding author.

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
