# Peer review of "Effect of Precipitates on the Mechanical Performance of 7005 Aluminum Alloy Plates"

_materials, 2022, doi:10.3390/ma15175951_

Round 1

Reviewer 1 Report

Dear Authors,

the publication in which you investigated the effect of precipitates on the conductivity and mechanical performance of a 7005 aluminum alloy plate shows the enormous amount of research you have done. However, not all measurement results have been interpreted, or their interpretation needs to be corrected. Despite many publications on a given subject, the authors refer to a small amount of it. Please refer to the comments below if you wish to publish an article in the Materials.

Comment#1:

Consider posting one series of photos from optical microscopy. I think that it is not necessary to include the same images only in different microscope modes. 

Comment#2:

Line 145:  You wrote: "No evident differences were found in the shape, size, quantity, and distribution of the micron-sized insoluble excess constituent particles α-Al(FeMn)Si between the alloy plates aged at 120 °C for different times, indicating that aging at 120 °C for less than 192 h had no significant effect on the configuration of the excess constituent particles in the 7005 aluminum alloy plates". Have you somehow analyzed the proportion and shape of the particles? How do you know that there are only α-Al(FeMn)Si particles in the image? You can use OM or SEM image analysis to determine the proportion and shape of the particles. Make EDX/SEM maps to determine particle composition, if possible.

Comment#3:

Fig. 4: Circle the precipitates in the TEM images because they are hard to see (for samples below 24h). Perhaps changing the contrast would help.

Comment#4:

Fig. 5 i, j, k: increase the SAED contrast because I do not see spots from the phases marked with arrows or keep them with a circle.

Comment#5:

Fig. 5 a, c, e, g: mark the red square from which the STEM image was taken.

Comment#6:

Line 207: "but the number of precipitates remain the same". Are you sure? How do you analyze the contributions of individual precipitates?

Comment#7:

You confirmed the presence of individual precipitates by performing SAED diffraction (Fig. 5). Additionally, perform EDX/TEM, which will facilitate the recognition of individual particles in the matrix.

Comment#8:

The results of conductivity are presented in section 3.2. However, there are no interpretations of these results. Perhaps the following articles will help with the interpretation:

1. https://doi.org/10.1016/j.msea.2018.11.111

2. https://doi.org/10.1016/S1003-6326(17)60261-9

3. https://doi.org/10.1002/adem.202001249

Comment#9:

The above articles should also help to improve the discussion. Additionally, they suggest that you familiarize yourself with the works:

1. https://doi.org/10.1016/j.jallcom.2019.02.324

2. Huan Zhao, Master thesis, "Segregation and precipitation at interfaces in a model Al-Zn-Mg-Cu alloy", http://publications.rwth-aachen.de/record/770602/files/770602.pdf

3. https://doi.org/10.1016/S1003-6326(11)61398-8

It seems to me that it is necessary to improve the discussion and to emphasize the novelty of this work in relation to the articles already published. After these corrections, we will gladly accept the manuscript for publication.

Reviewer 2 Report

I want to congratulate the authors for the manuscript titled as “Effect of Precipitates on the Mechanical Performance of 7005 Aluminum Alloy Plates”. The article is interesting and deserves the attention of readers. However, there are several points in the article that require further explanation.

·         Abstract: writing is too generalized, and it is too long especially for explanation about the material and method process. The main theme of this paper is not described in the abstract. Abstract section should be concisely reflected the content and summarize the problem, the method, the results, and the conclusions. The abstract needs to be improved. Demonstrate in the abstract novelty, practical significance.

·         In the introduction section, more literature paper have to be included to explain the subject better. The introduction section has been written beautifully but need to include recent published papers on 7xxx series Al alloys and precipitation effect on the overall performance of these groups. (Please refer to most relevant papers).

Some giving citations need to be check such as [1-3], as they may not provide the required information in a sentence. Besides, add scientific novelty and practical relevance. Add a clear purpose to the article. Please show the literature gaps demonstrating the presented study fills it. At the last paragraph of the introduction, please clearly show the general outline of the paper and show the importance of the study along with the main aim.

·         Please specify the material properties, company name and county in the experimental section. Please give some explanatory information about production and characterization steps. For example, what kind of production process is used for Al plates. If they are purchased from company as-received, please give some information or not please explain production process. Besides, why Keller reagent or other etcher is not used for etching process. As you would agree,  grain boundaries, size, shape, etc.. are not visible without etching process but authors claimed that the “The grain size and shape of all these plates are observed to be roughly the same”, please explain this statement. Also, authors said that they are observed different precipitations as a function of aging time. However, there is no XRD results to see precipitates, intermetallic or secondary phases encountered in TEM results. If possible please perform XRD analysis or discuss these results with recently published most relevant studies.  

·         Language used in the manuscript is generally satisfying. However, writers should pay more attention of singular / plural nouns. Also, they should control the spell check/ punctuation of words and sentences. Please check all manuscript for language and misspellings. Besides, the color of some texts are not proper, such as page 15 line 587, 598, 605 and page  1 line 6,7 and page 2 line 65.  Please revisit all manuscript and correct such inconsistencies. Also, some text in the inset of figure 6 are not readable, please enlarge the text font.

·         Indeed, there are an impressive amount of results. Especially, discussion part of the manuscript is well enough and highly satisfying. However, the conclusions section needs to improve with selected and highlighted main findings. In conclusion section, it is necessary to more clearly show the novelty of the article and the advantages of the proposed method. Add qualitative and quantitative results of your work. Please try to emphasize your novelty, put some quantifications, and comment on the limitations. This is a very common way to write conclusions for a learned academic journal. The conclusions should highlight the novelty and advance in understanding presented in the work.

·         There is a reference problem. First, your reference list contains any paper from “Materials” journal. Second, the reference list needs to be revised and 50% of the citations must be published in the last 5 years regarding of Aluminium alloys, 7xxx series, etc.. If your work is convenient for this journal’s context then there are many references from this or other journal.

-----------------------------------------------------------------------------------------------------------------

The article is interesting but needs to be improved. Authors should carefully study the comments and make improvements to the article step by step. After minor corrections/changes can an article be considered for publication in the "Materials".

Reviewer 3 Report

Manuscript is devoted to the study of the effect of precipitates on the mechanical properties of the 7005 aluminum alloy plate. The authors conducted tensile tests and fatigue tests, and also analyzed the microstructure of the alloy after artificial aging with different time. The authors analyzed and discussed in detail the results obtained and described the relationship between mechanical properties and the parameters of precipitates after artificial aging. In my opinion, the authors have done a large amount of experiments and obtained significant results. 

I have a few small comments on the manuscript:

1. Lines 420-422. “Therefore, the yield and tensile strengths of the alloy plates increased from 168 MPa and 255 MPa, respectively, after aging for 1 h to 174 MPa and 264 MPa, respectively, after aging for 4 h.” Figure 7b shows that, taking into account the accuracy of measurements up to 4 hours of aging, there is no significant change in the tensile strength.

2. Similar comment regarding line 547. “The distance between the fatigue striations in the fracture crack propagation zone of the alloy plates was 0.31 μm, slightly larger than the 0.28 μm in the fracture crack propagation zone of the alloy plates aged for 96 h.” What is the zone measurement accuracy? It seems to me that the measurement accuracy is greater than 0.31-0.28=0.03 μm.

3. Figure 8. Why don't the authors list confidence intervals along with average fatigue instead of values for each sample? Moreover, the figure indicates the designation for only one sample. It is also necessary to specify a unit of measurement for the time of aging. Moreover, only time without temperature can be indicated on the x-axis. I think the authors should correct this Figure.

Round 2

Reviewer 1 Report

Dear Author,

thank you for your responses. I think you should focus more on analyzing the individual particles in the future. Due to particle size, analysis of the chemical composition may be complicated, but it should be possible in the case of longer times. Please enter the text information about the software you used to analyze the particles and how many particles were taken to calculate their size.

Reviewer 3 Report

The authors responded to all comments and significantly revised the manuscript. I think the manuscript can be accepted for publication.

Author Response

Thank you very much for your recognition.